# Innovative Use of Sheep Wool for Obtaining Materials with Improved Sound-Absorbing Properties

**DOI:** 10.3390/ma13030694

**Published:** 2020-02-04

**Authors:** Simona Ioana Borlea (Mureşan), Ancuţa-Elena Tiuc, Ovidiu Nemeş, Horaţiu Vermeşan, Ovidiu Vasile

**Affiliations:** 1Faculty of Materials and Environmental Engineering, Technical University of Cluj-Napoca, Cluj-Napoca, 28 Memorandumului Street, 400114 Cluj-Napoca, Romaniahoratiu.vermesan@imadd.utcluj.ro (H.V.); 2National Institute for Research and Development in Environmental Protection, 294 Blvd Splaiul Independenței, Sector 6, 060031 Bucharest, Romania; 3Department of Mechanics, Politehnica University of Bucharest, 313 Splaiul Independentei, 060042 Bucharest, Romania; ovidiu.vasile@upb.ro

**Keywords:** sheep wool recovery, acoustic materials, sound absorption coefficient

## Abstract

In recent years, natural materials are becoming a valid alternative to traditional sound absorbers due to reduced production costs and environmental protection. This study explores alternative usage of sheep wool as a construction material with improved sound absorbing properties beyond its traditional application as a sound absorber in textile industry or using of waste wool in the textile industry as a raw material. The aim of this study was to obtain materials with improved sound-absorbing properties using sheep wool as a raw material. Seven materials were obtained by hot pressing (60 ÷ 80 °C and 0.05 ÷ 6 MPa) of wool fibers and one by cold pressing. Results showed that by simply hot pressing the wool, a different product was obtained, which could be processed and easily manipulated. The obtained materials had very good sound absorption properties, with acoustic absorption coefficient values of over 0.7 for the frequency range of 800 ÷ 3150 Hz. The results prove that sheep wool has a comparable sound absorption performance to mineral wool or recycled polyurethane foam.

## 1. Introduction

From a sustainable development perspective, an important goal is to choose raw materials that are easily recyclable and renewable as well as locally available and environmentally friendly. This includes timber, clay, stone, straw, bio-based fibers, and sheep wool, provided that any further processing is carried out with low energy consumption.

The origin of these materials can be vegetable or animal so that their manufacturing has a low environmental impact due to the energy saved in the production process [1]. The use of natural fibers as raw material for acoustic applications have been intensively studied [2,3,4,5,6,7,8,9,10,11,12,13,14], especially in recent years. Many industries are moving toward natural material-based, environmentally friendly products [15,16]. This may be due to the fact that the energy required to process these types of materials is lower compared to that required for synthetic materials. For instance, processing 1 m^3^ of sheep wool insulation produces almost 5.4 kg of CO_2_, whereas the quantity of CO_2_ produced is 135 kg in the case of mineral wool [17]. Thus, the environmental impact when using these types of materials is low, and there is no negative effect on the environment [18].

Many natural materials, such as bamboo, kenaf, sisal, flax, hemp, sheep wool, cork, or coconut fibers, show good sound-absorbing performance and can therefore be used as sound absorbers in acoustic rooms and noise barriers [19,20].

Sheep wool is an easily recyclable, easily renewable, and environmentally friendly source of raw material, which consists of 60% animal protein fibers, 10% fat, 15% moisture, 10% sheep sweat, and 5% impurities on average. Zach et al. evaluated the thermal, hygrothermal, and acoustic performance of samples of sheep wool materials. A mixture of sheep wool was mechanically fastened to a reinforcing cloth with varying thickness and density. The results showed that sheep wool was characterized by high hygroscopicity that reached up to 35% and that sheep wool could therefore be an excellent acoustic insulating material [21]. 

Del Rey et al. studied sheep wool as a sustainable material for acoustic applications. The materials were made from sheep wool by thermofusion with polyester fibers obtained from recycled polyethylene terephthalate (PET) flakes that acted as a binder (PET fibers melt at 140–150 °C). The final material had 80% sheep wool fibers (first quality, second quality, or blend), and the remaining 20% was PET fibers. From the measurement results, it was demonstrated that sheep wool with PET fiber was a good sound-absorbing material at medium and high frequencies, with acoustic absorption coefficient values of over 0.5 for the frequency range of 600 ÷ 3150 Hz for the best material obtained [1].

Until now, sheep wool has traditionally been used in the textile industry for the manufacturing of conventional woolen products, such as carpets, garments, curtains, covers, and bedding. More recently, they have also been used in the building industry due to their thermal properties. For the fabrication of wool-based building materials, coarse fibers or those fibers that cannot be used in the textile industry are generally used [22]. Wool has good thermal characteristics, with the thermal conductivity of wool panels varying between 0.040 and 0.041 W/mK for densities of 25 ÷ 92.5 kg/m^3^ [23].

Sheep wool fibers have a similar size as mineral fibers. A 33 ÷ 36 µm sheep wool fiber would roughly be the same size as PET polyester fibers (33 µm) [24] or Kenaf fibers (36 µm) [25]. Unlike synthetic fibers, sheep fibers do not have a fixed thickness. Their thickness range has a standard deviation of 2 µm, according to scientific literature [26]. The fiber diameter also depends on the breed of the sheep.

The surface of wool fibers has many scales, and the fibers can only move on one direction. Under mechanical agitation, friction, and pressure in the presence of moisture and heat, the scale edge of one fiber locks into the interscale gap of another fiber like a “ratchet” mechanism. The fibers interlock and cannot return to their original positions, resulting in irreversible felting shrinkage [27].

Pressed felt is produced from wool or animal hair by mechanical agitation and compression of the fibers in warm, moist conditions [28].

The aim of this study was to analyze the sound absorption coefficient of some materials or structures based on sheep wool as an alternative to the classical (wool felts, mineral wool, or foams) or the new series of improved sound absorbers. The sound absorption capability of sheep wool was measured in an impedance tube. Experimental results indicated the material’s excellent performance in the development of building elements for sound absorption with or without the addition of other elements (polyurethane foam, epoxy, or polyester resin).

Compared with the classical acoustic materials existing in the market or in the literature, the ones obtained in this research have the advantage of good properties. They are also environmentally friendly due to the fact that no binders are used, and the working parameters (pressure and temperature) require low energy consumption. These materials with very good acoustic absorption properties can be obtained by hot pressing without the presence of humidity compared to the standard mode of felting.

## 2. Materials and Methods

### 2.1. Materials

In order to obtain the desired sound-absorbing materials, black merino sheep wool (different shades of black, including dark brown) was used. Figure 1 shows the raw sheep wool (60 ÷ 80 mm length, 18 ÷ 20 µm fineness, ripple of 100 mm, and density of 3.4578 g/cm^3^). Prior to experiments, the raw wool was washed to remove impurities, sand, and dust, and it was then dried and carded.

### 2.2. Manufacturing Process

The samples used in this research were obtained by hot and cold pressing. The mold used to obtain the material samples had a cylindrical shape with two aluminum hot plates (top and bottom). The samples were heated from both sides to obtain a uniform temperature in the mold. The mold was equipped with four thermocouples disposed on the outside, which were connected to a temperature regulator and measured the working temperature. The mold was also fitted with a thermostat to maintain a constant temperature. Figure 2 shows the mold that was used to obtain the material samples.

Because the ability of a material to reduce the acoustic energy depends on its thickness, different wool quantities were considered in order to prepare different sample thicknesses [29].

The mold for the cold-pressed samples was made of steel in rectangular shape with a lid. To obtain the hot-pressed samples, hot pressing was done in a mold by applying a pressure of 0.05 ÷ 6 MPa on the material, which was heated to 60 ÷ 80 °C. When wool fibers are heated, they easily fill the new shape, and the pressure in the mold forces the wool fibers to compress. The obtained samples were kept in the mold under pressure until complete cooling. After this, the mold was opened, and the sample was extracted. The main parameters that were followed for this process were pressure, pressing time, and temperature. The required heat was transferred through the mold walls. The required pressure force was obtained from a manually operated hydraulic press (Unicraft WPP 50 E, Stürmer Machinen Gmbh, Hallstadt, Germany).

Eight sound-absorbing materials were obtained following the procedure described above. The obtained materials were divided into three groups (Figure 3) depending on the embodiment: hot-pressed wool moistened with water (WHW, labelled as A), hot-pressed wool (WH, labelled as B), and cold-pressed wool (WC, labelled as C). Table 1 presents the technical parameters of the obtained material samples.

The materials in group A were made of wool by hot pressing (pression 3 and 6 MPa and temperature 60, 70, and 80 °C). In order to analyze the influence of humidity on the new materials, three tests were carried out by varying the amount of water used for moistening, i.e., 25, 50, and 75 mL. Results showed that the wool became plastic by wetting, especially at temperatures around 80 °C. It is upon this thermoplastic property that the pressing and elimination of the wrinkles in the wool fiber is based [31]. Thus, sample WHW80_6_75 heated at 80 °C had the consistency of a plywood with portions of glossy faces.

In the case of materials from group B, the wool was not wetted, and the initial wool layer (120 and 240 mm) was hot pressed at 4 and 0.05 MPa at a temperature of 80 °C. Four samples were obtained, the parameters and codes of which are presented in Table 1.

The material constituting group C was obtained by cold pressing a wool layer with an initial height of 40 mm at a pressure of 0.003 MPa.

### 2.3. Methods

The determination of apparent density was performed according to ISO 845:2006 [32]. The tested samples were cylindrical with a diameter of 63.5 mm and a specific height for each material. The dimensions of the specimens were measured with a 0.1 mm accuracy. The weight was determined using a laboratory balance with 0.01 g accuracy. The apparent density of specimens was calculated using the following formula: (1)$$\rho =\frac{m}{V}\;[g/{\mathrm{cm}}^{3}]$$
where *m* is the mass of the sample, and *V* is the volume of the sample.

The sound absorption coefficient at normal incidence (α) is the quotient between the acoustic energy absorbed by the surface of the test sample and the incident acoustic energy for a plane acoustic wave at normal incidence. ISO 10534-2 standard [33] establishes a test procedure to determine the sound absorption coefficient for normal incidence of acoustic absorbers by means of an impedance tube, two microphone positions, and a digital analysis system signal.

The method of measuring the acoustic absorption coefficient by means of the impedance tube is based on the fact that the reflection coefficient at normal incidence (r) can be calculated from the measured transfer function (H12) between two positions of the microphone at different distances from the sample. The transfer function of the incident (*H_I_*) and reflecting waves (*H_R_*) between the microphone positions are defined as follows [34]:(2)$$H_{I}=\frac{p_{2I}}{p_{1I}}=e^{-jk\cdot \left(x_{1}-x_{2}\right)}=e^{-jk\cdot s}$$
(3)$$H_{R}=\frac{p_{2R}}{p_{1R}}=e^{-jk\cdot \left(x_{1}-x_{2}\right)}=e^{-jk\cdot s}$$
where *s* is the distance between the two microphone positions; *x*_1_ and *x*_2_ are the distances from the reference point to microphone position 1 and 2, respectively; *p_I_* and *p_R_* are the sound pressure propagating in the incident and reflected direction, respectively; and *jk* is a complex-valued wavenumber.

The transfer function (*H*_12_) for the total sound field can be calculated with the following formula [34]:(4)$$H_{12}=\frac{p_{2}}{p_{1}}=e^{-jk\cdot \left(x_{1}-x_{2}\right)}=e^{-jk\cdot s}$$

The reflection coefficient (*r*) at the sample surface (*x* = 0) is as follows [33]:(5)$$r=\frac{H_{12}-H_{I}}{H_{R}-H_{12}}e^{2jk\cdot x_{1}}$$

The sound absorption coefficient at normal incidence is calculated with the following formula [34]:(6)$$r=1-{\left|r\right|}^{2}$$

The α values were ascertained by producing standing waves in a tube with 63.5 mm diameter, so the tests were performed on circular samples (Figure 4) with a diameter of 63.5 mm. The circular samples were placed at the end of the Kundt’s tube Brüel&Kjaer Type 4206 during each test. Measurements were recorded at the third-octave frequency band within the intervals of 100 ÷ 3200 Hz and conducted at an air temperature of 26 °C, relative humidity of 55%, and pressure of 100.5 kPa.

## 3. Results and Discussion

### 3.1. Results for Apparent Density Tests

The material density is an important factor for the acoustic absorption of a material. As the density of the material increases, the sound absorption at medium and high frequencies also increases. Increased number of fibers per unit area increases the apparent density. Energy loss increases with the increase in friction surface, thus increasing the sound absorption coefficient [35].

The apparent density determined for the obtained materials is shown in Figure 5. It can be seen that the hot-pressed materials had a much higher density than the cold-pressed materials. The density of the materials made from sheep wool increased with the increase in pressure.

### 3.2. Results of Acoustic Tests

The acoustic characterization of the materials was based on the sound absorption coefficient α [36]. This parameter is the ratio of absorbed sound intensity to incident sound intensity on a surface [37]. The potential for materials to absorb sound energy depends on the following factors: density, thickness, porosity, fiber diameter, airflow resistivity, tortuosity, surface impedance, compression, air gap, and multilayers [19,38].

#### 3.2.1. The Effect of Material Thickness on the Sound Absorption Coefficient

This section highlights and discusses the variation in acoustic absorption coefficient with the thickness of materials obtained from hot-pressed sheep wool at 80 °C and a pressure between 4 and 0.05 MPa. The influence of material thickness obtained from compressed sheep wool at a pressure of 4 MPa on the coefficient of acoustic absorption is presented in Figure 6.

From Figure 6, it can be observed that material WH240_4 with 25 mm thickness had an acoustic absorption coefficient greater than WH120_4, which had a smaller thickness (15 mm), over the entire analyzed frequency range.

The thickness of the compressed materials at 0.05 MPa was 35 mm for material WH120_0.05 and 50 mm for WH240_0.05. 

The influence of compressed material thickness on the acoustic properties is shown in Figure 7. It can be observed that the material with the greatest thickness had the highest sound absorption coefficient values over the entire analyzed frequency range.

The sound absorption coefficient improved by increasing the composite thickness, an aspect that has been demonstrated in the literature [39]. Experimental testing performed on materials such as fiber felts, glass wool, paddy straw, textile waste, rubber crumbs, and polyester have all shown an increase in sound absorption with an increase in material thickness, especially at lower frequencies [1,40,41,42,43]. 

An analysis of the sound absorption coefficient values of our hot-pressed, wool-based material (Figure 6 and Figure 7) relative to other wool-based materials reported in the literature showed that the coefficient values were better at comparable thicknesses. The acoustic absorption coefficient values of the obtained materials at the frequency of 1000 Hz were 0.4 for WH120_4 (15 mm), 0.59 for WH240_4 (25 mm), and 0.84 for WH240_0.05 (50 mm). For other materials made from sheep’s wool [21], the coefficient values for different material thicknesses were 0.331 for 20 mm, 0.415 for 30 mm, and 0.7 for 40 mm.

#### 3.2.2. The Influence of Wool Compression on the Sound Absorption Coefficient

The studied materials were compressed at 0.05 and 4 MPa starting from an initial height of 240 mm and 120 mm, respectively, and the influence of the compression of wool fibers on the acoustic absorption coefficient is shown in Figure 8.

The compressed material WH240_0.05 had better sound-absorbing properties at 0.05 MPa with 0.01 g/cm^3^ density than the compressed material WH240_4 at 4 MPa with 0.61 g/cm^3^ density. For the materials marked as WH120_0.05 and WH120_4, the acoustic absorption coefficient had better values in the frequency range of 50 ÷ 1100 Hz and 1600 ÷ 3200 Hz compared to the compressed material at a lower pressure (WH120_0.05). This can be explained by the fact that the fibers within the material are brought closer to each other during compression. Thus, the material becomes more compact, the open porosity decreases, and the compression leads to a decrease in the thickness of the material [44].

The acoustic absorption properties of fibrous mat decrease during compression because the material thickness decreases during compression. Compression tests done on polyester fiber showed a drop in the absorption coefficient when the fibrous mat was compressed [45]. Fouladi et al. and Nor et al. they stated that compression affects the physical parameters of materials, including the flow resistivity, tortuosity, and porosity. These parameters define the link between the acoustic medium and the matrix [46,47].

#### 3.2.3. The Influence of the Presence of Water on the Sound Absorption Coefficient

The variation in the sound absorption coefficient depending on the amount of water used to obtain WHW80_6_50 and WHW80_6_75 materials is shown in Figure 9. The sound-absorbing properties of the obtained material by wetting with 50 mL of water (WHW80_6_50) were better than that of the one obtained using 75 mL of water (WHW80_6_75).

The decrease in the absorption coefficient values at frequencies below 2850 Hz for material WHW80_6_75 was due to the change in the thermoplastic properties of wool in the presence of water at 80 °C. On the material’s surface, plasticized and glossy areas appeared, which reflected the sound wave and did not allow it to penetrate the material for decreased sound intensity [31].

Polypropylene/jute webs with a thickness of 4.28 mm and a density of 0.65 g/cm^3^ were found to have an acoustic absorption coefficient α < 0.2 in the frequency range of 100 ÷ 1600 Hz [4], while WHW80_6_50 (2.5 mm) and WHW80_6_75 (3 mm) with a density of 0.7 g/cm^3^ and 0.71 g/cm^3^ had a sound absorption coefficient of α < 0.63 in the frequency range of 100 ÷ 1600 Hz.

#### 3.2.4. Influence of Cold/Hot Compression on the Sound Absorption Coefficient

The influence of the compression mode of the wool fibers (cold or hot) on the acoustic absorption coefficient is highlighted in Figure 10. It can be observed that the materials obtained by hot pressing (WH240_4 and WH120_4) had superior sound-absorbing properties compared to the material obtained by cold pressing (WC40). 

Considering the density of the hot-pressed materials (0.61 ÷ 0.65 g/cm^3^) was higher than the density of the cold-pressed materials (WC40 0.02 g/cm^3^), it can be said that the sound absorption coefficient at high and medium frequencies is higher for materials with higher density [48]. It can be seen from Figure 10 that sample WH240_4 with a density of 0.61 g/cm^3^ and a thickness of 25 mm had the best acoustic absorption coefficient values for the entire frequency range analyzed, reaching a maximum of 0.91 at 2500 Hz. In comparison, sample WC40 with a density of 0.02 g/cm^3^ and a thickness of 25 mm had much lower absorption coefficient values, barely reaching 0.3 in the frequency range of 2500 ÷ 3150 Hz. 

#### 3.2.5. Comparisons with Other Materials

In order to accentuate the sound-absorbing properties of the materials obtained in this research, a comparative study with other materials in the market (rigid polyurethane foam 40 mm, flexible polyurethane foam 40 mm, and mineral wool 50 mm) and a material from the literature (sheep wool mechanically fastened on cloth [21]) was carried out. The results obtained are shown in Figure 11. It can be observed that the obtained material WH240_0.05 (sheep wool hot pressed at 80 °C with 0.05 MPa) had the best sound-absorbing properties at frequencies below 2000 Hz, while it had values almost identical to mineral wool in the frequency range of 2000 ÷ 3200 Hz.

Compared to the material obtained by Zach et al. [21], also from sheep wool, material WH240_0.05 obtained in this research had better acoustic absorption properties at frequencies higher than 315 Hz. 

The material with 80% sheep wool fibers (40% first quality and 40% second quality) and 20% PET fibers (50 mm) with an acoustic absorption coefficient of more than 0.6 for the frequency range of 800 ÷ 3150 Hz [1] had lower acoustic properties compared to WH240_0.05 (50 mm) with an acoustic absorption coefficient of more than 0.72 for the frequency range of 800 ÷ 3150 Hz.

At frequencies greater than 1200 Hz, material WH240_4 with 20 mm thickness had an absorption coefficient value greater than the flexible polyurethane foam with 40 mm thickness. It should be mentioned that the flexible polyurethane foam maintained the best sound-absorbing properties at frequencies below 400 Hz. 

## 4. Conclusions

Obtaining environmentally friendly materials with very good acoustic properties from natural and renewable raw materials, such as sheep wool without using any binder, is an important step in solving environmental problems and, at the same time, finding new methods of using wool. By simply hot pressing wool, a material that can be processed and manipulated can be obtained.

Hot-pressed materials have a much higher density than cold-pressed materials. The density of materials made from hot-pressed sheep wool increases with increasing pressure.

In this research, material WH240_0.05, which had a 240 mm layer of wool and 50 mm thickness and was hot-pressed at 80 °C and 0.05 MPa, had higher sound absorption coefficient values over the entire analyzed frequency range compared to WH120_0.05, which was obtained under the same conditions but with a smaller thickness and a 120 mm layer of wool.

Due to the thermoplastic properties of wool in the presence of water at a temperature of 80 °C, the sound absorption coefficient of material WHW80_6_75 had lower values at frequencies lower than 2850 Hz compared to the material with a lower water content.

WH240_0.05, which had 0.01 g/cm^3^ density and was pressed at 0.05 MPa, had better sound-absorbing properties than WH240_4, which was pressed at 4 MPa and had 0.61 g/cm^3^ density. During the compression, the fibers of materials come close, so the open porosity decreases and the compression increases.

The WH240_0.05 material obtained in this study had the best sound-absorbing properties at frequencies below 2000 Hz, while it had values almost identical to mineral wool in the frequency range of 2000 ÷ 3200 Hz. Thus, hot-pressed sheep wool has better or at least equal sound-absorbing properties as that of mineral wool, which is one of the most widely used sound-absorbing fibrous materials.

The field of use for the obtained materials is wide, but other characteristics will have to be determined.

## Figures and Tables

**Figure 1 materials-13-00694-f001:**
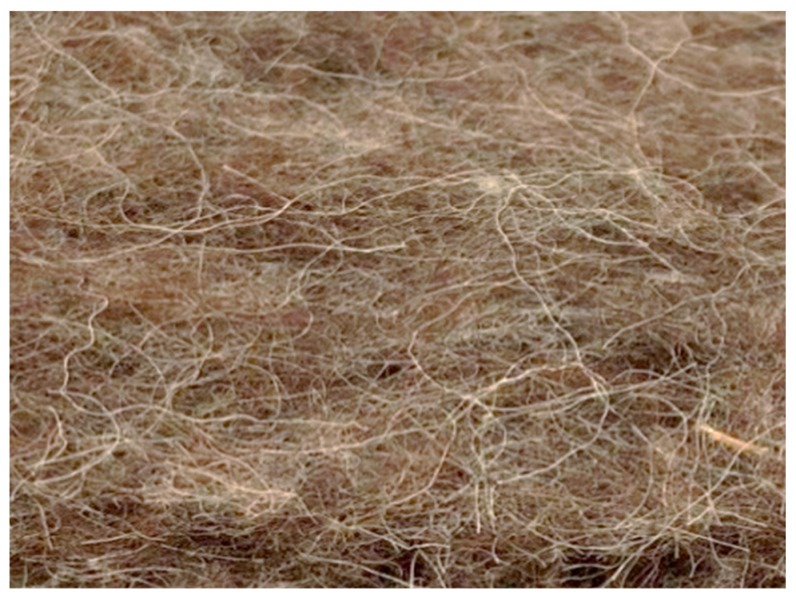
The raw material used: black merino sheep wool.

**Figure 2 materials-13-00694-f002:**
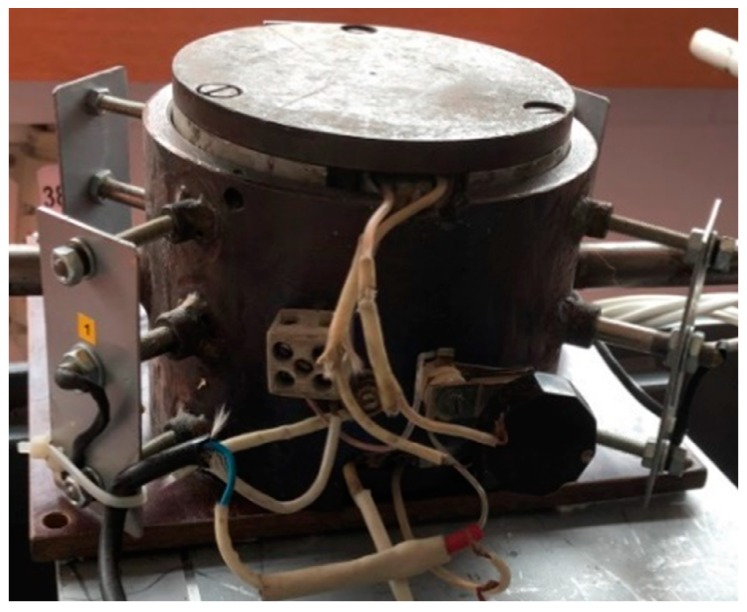
The mold used to obtain material samples by hot pressing [30].

**Figure 3 materials-13-00694-f003:**
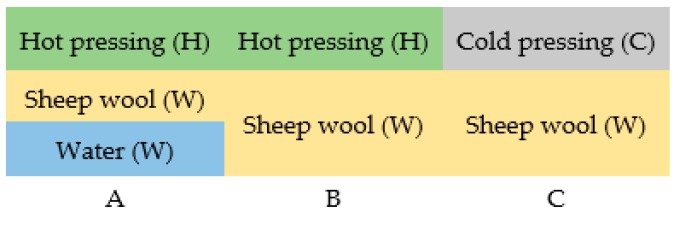
Groups of obtained materials.

**Figure 4 materials-13-00694-f004:**
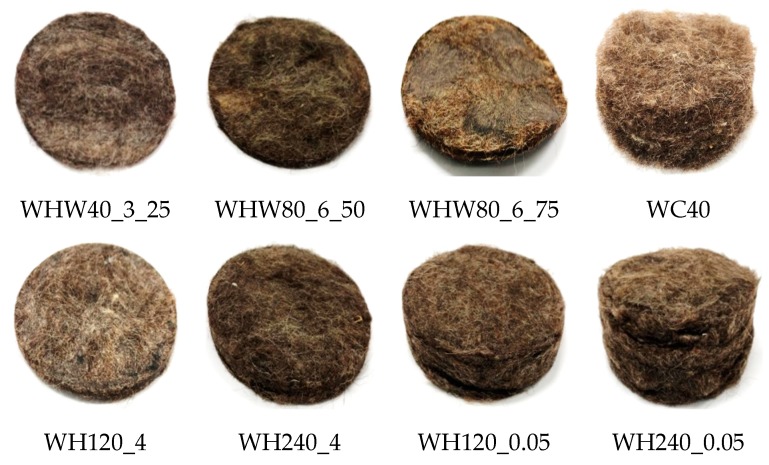
Samples prepared for sound absorption coefficient measurement.

**Figure 5 materials-13-00694-f005:**
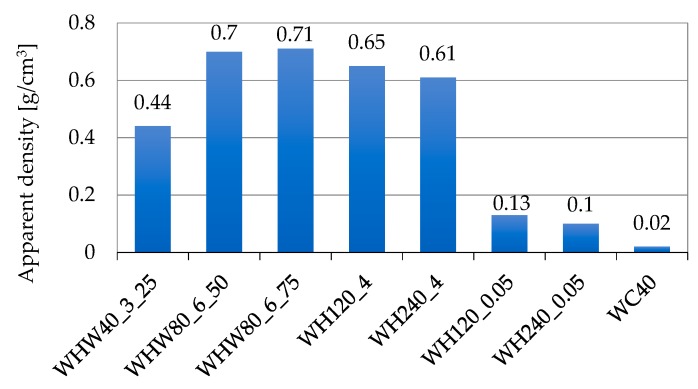
Apparent density values.

**Figure 6 materials-13-00694-f006:**
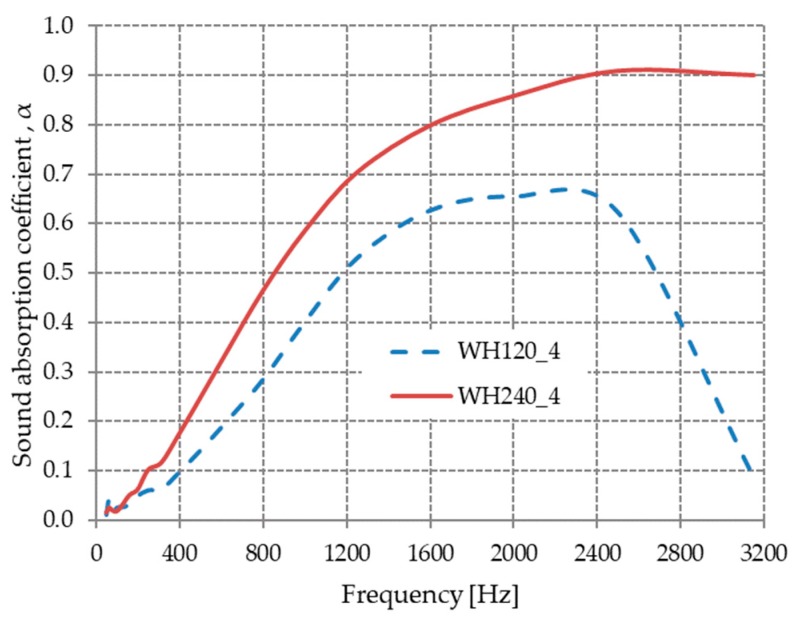
Variation in the acoustic absorption coefficient with the material thickness for WH120_4 and WH240_4.

**Figure 7 materials-13-00694-f007:**
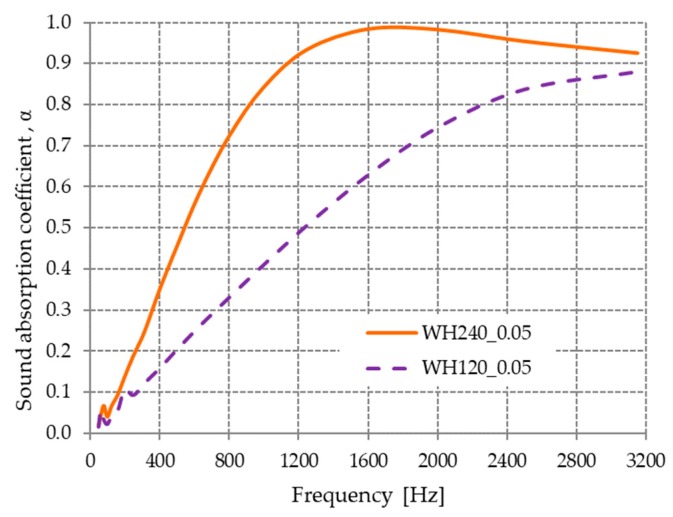
Variation in the acoustic absorption coefficient with the material thickness for WH120_0.05 and WH240_0.05.

**Figure 8 materials-13-00694-f008:**
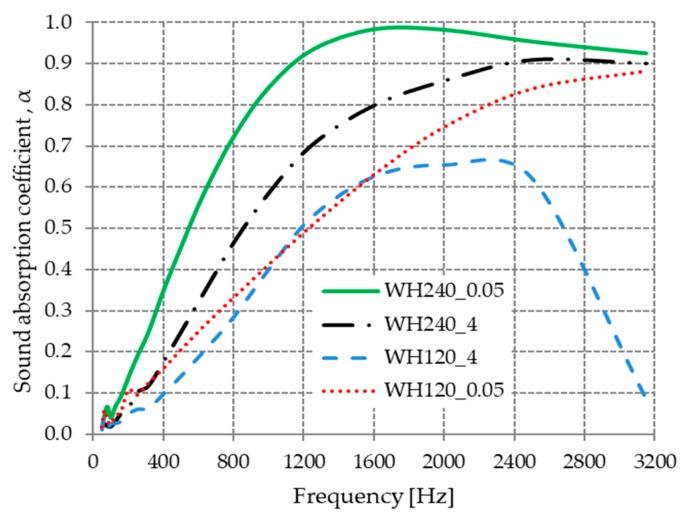
Variation in the acoustic absorption coefficient with the compaction pressure.

**Figure 9 materials-13-00694-f009:**
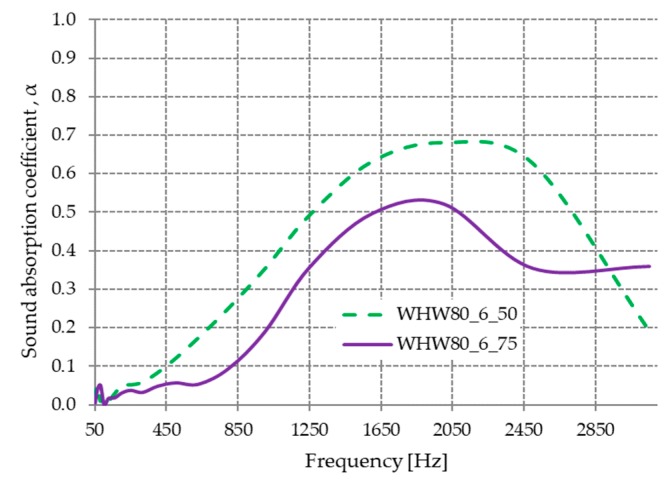
Variation in the acoustic absorption coefficient with the quantity of water.

**Figure 10 materials-13-00694-f010:**
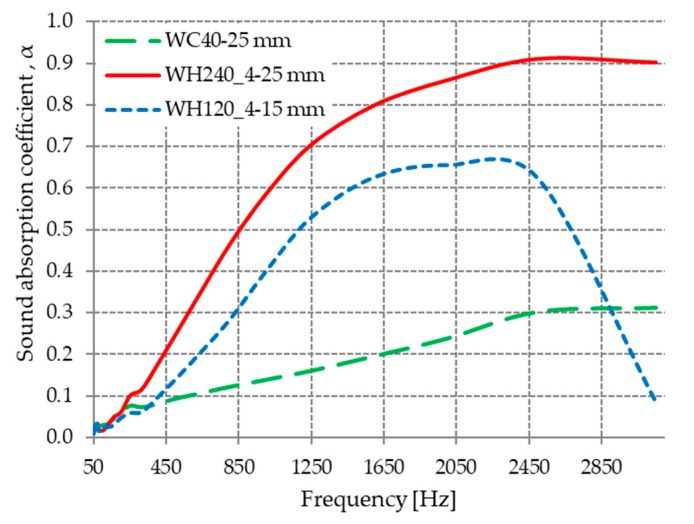
Variation in the acoustic absorption coefficient with frequency.

**Figure 11 materials-13-00694-f011:**
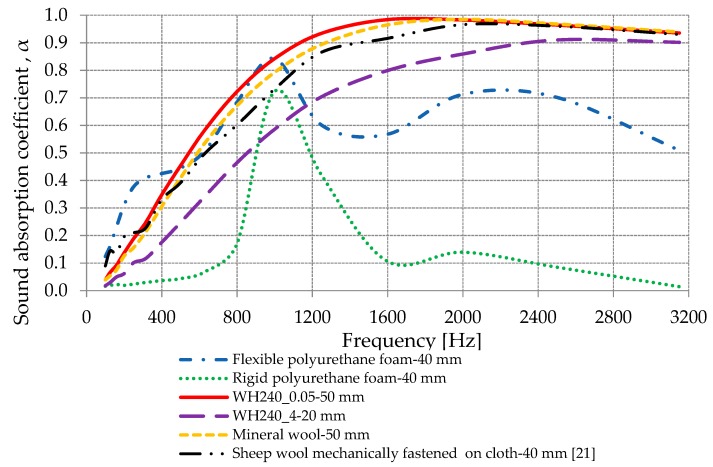
Variation in the acoustic absorption coefficient for WH240_0.05, WH240_4, flexible polyurethane foam, rigid polyurethane foam, and mineral wool.

**Table 1 materials-13-00694-t001:** Technical parameters of the materials obtained.

Group	Code	Initial Height (mm)	Final Height (mm)	Temperature (°C)	Pressure (MPa)	Water (ml)
A	WHW40_3_25	40	1	60	3	25
	WHW80_6_50	80	2.5	70	6	50
	WHW80_6_75	80	3	80	6	75
B	WH120_4	120	15	80	4	-
	WH240_4	240	25	80	4	-
	WH120_0.05	124	35	80	0.05	-
	WH240_0.05	240	50	80	0.05	-
C	WC40	40	25	25	0.003	-

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
