# Peer review of "Innovative Use of Sheep Wool for Obtaining Materials with Improved Sound-Absorbing Properties"

_materials, 2020, doi:10.3390/ma13030694_

Round 1

Reviewer 1 Report

Dear authors,

I am glad that you tried to utilise the natural material - sheep wool, because in the past years , the utilisation of this material was very low. Also this material can replace the plastics in some places. Also in automotive industry  or civil engineering the natural materials can be sucessfully used.

You have described the production of the testing samples and results of measuring. But I recommend to complete  and  to add some more information about acoustic measuring appararus and its technical parameters.

Author Response

Dear reviewer thank you for your time and your analyse of our work.

We have added more information about acoustic measuring apparatus in section 2.3

Reviewer 2 Report

The main problem I have with this paper is that it claims to be an innovative investigation.   Wool felts have been tested and used for acoustic absorption for at least the last 60 years.  There is much literature on evaluating felts for noise absorption and this literature should be cited and the present research related to it.  Felts are available commercially for noise reduction with specified characterisitics.

The research may be valid and it use demonstrated but it cannot be justified or amplified as research on a brand new material.  That is deceptive.

The paper should be proofed and the English improved but detailed comments at this time are not called for until the basic problem is resolved.

Author Response

Dear reviewer thank you for your time and your analyse of our work.

1. The main problem I have with this paper is that it claims to be an innovative investigation.   Wool felts have been tested and used for acoustic absorption for at least the last 60 years.  There is much literature on evaluating felts for noise absorption and this literature should be cited and the present research related to it.  Felts are available commercially for noise reduction with specified characteristics.

Dear reviewer we have added more references. Reference 2, 3, 17 and 18 to complete the state of the art in the field.

We didn't obtain felts. In this work we present samples of new structures based on sheep wool

2. The research may be valid and it use demonstrated but it cannot be justified or amplified as research on a brand new material.  That is deceptive.

The aim of our research activity is to obtain and characterize plates or rigid structures to develop building sound absorption elements with or without addition of other elements (polyurethane foam, epoxy or polyester resin)

3. The paper should be proofed and the English improved but detailed comments at this time are not called for until the basic problem is resolved.

We have improved the English. 

Reviewer 3 Report

The paper is not suitable for publication as is due to some problems, but it can be improved if the authors are available to work on it.

1. The english is not good: often it is very hard to understand what the authors want to say. 

2. All the figures captions are not clear.

3. the authors do not describe the method used to measure the apparent density of their samples: they just say that they did the measurement according to ISO 845-2006.

4. the authors should improve the description of the experimental methods adopted in their experiments in order to improve the scientific  quality of their paper.

5. the authors measure the apparent density and sound attenuation of their samples but they do not write any formula!!!

6. the conclusions should clearly describe the obtained results and show some comments related to the potential applications of their research and the improvements they obtained with respect to the current research.

Author Response

Dear reviewer thank you for your time and your analyse of our work.

1. The english is not good: often it is very hard to understand what the authors want to say. 

Dear reviewer we have improve the the English editing

2. All the figures captions are not clear.

We have improved the quality of the figures

3. the authors do not describe the method used to measure the apparent density of their samples: they just say that they did the measurement according to ISO 845-2006.

We have describe better the method in section 2.3 Methods

4. the authors should improve the description of the experimental methods adopted in their experiments in order to improve the scientific quality of their paper.

We improve the description of the experimental method

5. the authors measure the apparent density and sound attenuation of their samples but they do not write any formula!!!

We wrote the formula - (1) - in 2.3. Methods

6. the conclusions should clearly describe the obtained results and show some comments related to the potential applications of their research and the improvements they obtained with respect to the current research.

Round 2

Reviewer 2 Report

Look up the definition/description of felt.  It includes compressed wool fibers.  Such felt is sold commercially for sound absorbing purposes.  Therefore, the claim that you are investigating a new material is completely wrong and deceptive.

I know that it is desirable to do new and innovative work, but ignoring the past and present doesn't make work innovative.

I do not want to write your paper for you but:  the only way this work is acceptable is to admit felt has a long history of use and has been recognized as a good sound absorber for many decades.

With an honest introduction, you can then discuss any differences in fabrication and how you are optimizing the material for use as an absorber.  But, you must also research the literature to determine if you are doing anything new.  Since felt is available commercially, I suspect it has been the subject of optimization.

Author Response

Look up the definition/description of felt.  It includes compressed wool fibers.  Such felt is sold commercially for sound absorbing purposes.  Therefore, the claim that you are investigating a new material is completely wrong and deceptive.

I know that it is desirable to do new and innovative work, but ignoring the past and present doesn't make work innovative.

I do not want to write your paper for you but:  the only way this work is acceptable is to admit felt has a long history of use and has been recognized as a good sound absorber for many decades.

With an honest introduction, you can then discuss any differences in fabrication and how you are optimizing the material for use as an absorber.  But, you must also research the literature to determine if you are doing anything new.  Since felt is available commercially, I suspect it has been the subject of optimization.

Dear reviewer

first of all we don't want to ignore or to minimalise the work of other researchers.

We have change the title of our manuscript. 

We have improve the "introduction" section and we hope that it is allright now.

Reviewer 3 Report

the authors corrected their paper according to all the comments I added in the report.

the english could be still improved but I believe that it is sufficiently clear.

I think that the paper can be published.

I really hope that the authors will continue to work in this field and never stop the research of new type and shape of sound absorbing materials. 

good work

Author Response

Dear reviewer thank you for your time and your analyse of our work. We appreciate yours remarks